# The inability to walk unassisted at hospital admission as a valuable triage tool to predict hospital mortality in Rwandese patients with suspected infection

Arthur Kwizera[1], Olivier Urayeneza[2,3], Pierre Mujyarugamba[2], Jens Meier[4], Andrew J. Patterson[5], Lori Harmon[6], Joseph C. Farmer[7], Martin W. Dünser[4]*, for the "Sepsis in Resource-Limited Nations" Task Force of the Surviving Sepsis Campaign[¶]

1 Department of Anaesthesia and Critical Care, Makerere University College of Health Sciences, Kampala, Uganda, 2 Gitwe Hospital and Gitwe School of Medicine, Gitwe, Rwanda, 3 Department of Surgery, California Medical Center, Los Angeles, CA, United States of America, 4 Department of Anesthesiology and Intensive Care Medicine, Kepler University Hospital and Johannes Kepler University Linz, Linz, Austria, 5 Department of Anesthesiology, Emory University, Atlanta, Georgia, United States of America, 6 Society of Critical Care Medicine on behalf of the Surviving Sepsis Campaign, Mount Prospect, IL, United States of America, 7 Department of Critical Care Medicine, Mayo Clinic, Phoenix, AZ, United States of America

¶ The complete membership of the author group can be found in the Acknowledgments.
* Martin.Duenser@kepleruniklinikum.at

**Data Availability Statement:** Data cannot be shared publicly because of ownership rights. Data are made available from the Surviving Sepsis

## Abstract

### Objective

To assess the value of the inability to walk unassisted to predict hospital mortality in patients with suspected infection in a resource-limited setting.

### Methods

This is a *post hoc* study of a prospective trial performed in rural Rwanda. Patients hospitalized because of a suspected acute infection and who were able to walk unassisted before this disease episode were included. At hospital presentation, the walking status was graded into: 1) can walk unassisted, 2) can walk assisted only, 3) cannot walk. Receiver operating characteristic (ROC) analyses and two-by-two tables were used to determine the sensitivity, specificity, negative and positive predictive values of the inability to walk unassisted to predict in-hospital death.

### Results

One-thousand-sixty-nine patients were included. Two-hundred-one (18.8%), 315 (29.5%), and 553 (51.7%) subjects could walk unassisted, walk assisted or not walk, respectively. Their hospital mortality was 0%, 3.8% and 6.3%, respectively. The inability to walk unassisted had a low specificity (20%) but was 100% sensitive (CI95%, 90–100%) to predict in-hospital death (p = 0.00007). The value of the inability to walk unassisted to predict in-hospital mortality (AUC ROC, 0.636; CI95%, 0.564–0.707) was comparable to that of the qSOFA score (AUC ROC, 0.622; CI95% 0.524–0.728). Fifteen (7.5%), 34 (10.8%) and 167 (30.2%)

Campaign for medical researchers upon request. Please contact ssc@sccm.org.

**Funding:** This study was funded by the Life Priority Fund, the Hellman Foundation and the King Baudouin Foundation. The research project was supported by the European Society of Intensive Care Medicine and the Society of Critical Care Medicine through the Surviving Sepsis Campaign. Arthur Kwizera is supported by DELTAS Africa Initiative grant #DEL-15-011 to THRiVE-2. The DELTAS Africa Initiative is an independent funding scheme of the African Academy of Sciences (AAS)'s Alliance for Accelerating Excellence in Science in Africa (AESA) and supported by the New Partnership for Africa's Development Planning and Coordinating Agency (NEPAD Agency) with funding from the Wellcome Trust grant #107742/Z/15/Z and the UK government. The views expressed in this publication are those of the author(s) and not necessarily those of AAS, NEPAD Agency, Wellcome Trust or the UK government. The funders had no role in study design, data collection and analysis, decision to publish, or preparation of the manuscript.

**Competing interests:** The authors have declared that no competing interests exist.

patients who could walk unassisted, walk assisted or not walk presented with a qSOFA score count ≥2 points, respectively (p<0.001). The inability to walk unassisted correlated with the presence of risk factors for death and danger signs, vital parameters, laboratory values, length of hospital stay, and costs of care.

## Conclusions

Our results suggest that the inability to walk unassisted at hospital admission is a highly sensitive predictor of in-hospital mortality in Rwandese patients with a suspected acute infection. The walking status at hospital admission appears to be a crude indicator of disease severity.

## Introduction

Infectious diseases and sepsis are leading causes of death in resource-limited nations [1,2]. Unfortunately, in a resource-limited environment differentiating between patients with significant infection who will develop life-threatening sepsis complications versus patients who will manifest serious infection only is difficult. Furthermore, lack of routine diagnostic technology hampers the identification of patients with the highest risk of developing life-threatening sepsis [1,3]. We aimed to seek an additional assessment methodology that is simple, directly applicable to this patient population, and easy to accomplish and interpret. Longer term, it is our goal to translate improved identification of patients at hospital admission with the highest risk of death from infection into increased clinical support where appropriate.

In a recent prospective, before-and-after feasibility trial, our working group showed that a focused education program and implementation of an infection treatment bundle in clinical practice increased the rate of early evidence-based interventions in patients with acute infections (mostly malaria) admitted to a Sub-Saharan African district hospital [4]. In view of the fact that all patients who were hospitalized with an acute infection were included into the study, the overall disease severity and mortality (3.8%) was low [4]. In order to reduce the death toll from infectious diseases in this setting, future research efforts should focus primarily on patients with an acute infection who are at risk of in-hospital death. While some modified illness scores have shown promise [5,6], no triage tool exists which could early and reliably differentiate between patients with a minimal risk of death and patients who are at risk of dying from their infectious process.

In this analysis, we assessed the value of the inability to walk unassisted at hospital presentation to predict in-hospital mortality in patients with suspected infection hospitalized at the Gitwe District Hospital in rural Rwanda. We hypothesized that this simple prognostic indicator of disease severity accurately portended in-hospital death.

## Materials and methods

This is a *post hoc* analysis of an investigator-initiated, single-center, prospective, before-and-after feasibility trial, which was conducted at the adult and pediatric emergency department of the Gitwe District Hospital in Rwanda between March 2016 and March 2017 [4]. The trial protocol was pre-published (www.clinicaltrials.gov; NCT02697513), reviewed and approved by the National Health and Research Committee as well as the Rwanda National Ethics

Committee (No. 007/RNEC/2016). Written informed consent was obtained from all study patients or parents/next of kin in children.

## Setting

Details about the study site were previously published [4]. Briefly, the Gitwe District Hospital serves a population of approximately 160,000, has 200 beds and one emergency department with a separate adult and a pediatric admission room. No intensive care or high dependency unit is available, and supporting facilities such as the laboratory or radiology department are only modestly equipped. The hospital laboratory cannot provide chemical analyses to systematically evaluate organ functions and cannot process microbiological samples. Patients usually reach the hospital by foot, private cars, or when referred from regional healthcare centers, by the hospital's own ambulance.

## Study patients

All patients who were hospitalized because of a suspected acute infection were enrolled. Patients who were not able to walk unassisted before this disease episode (e.g. because of young age, neurological conditions, severe chronic disease or frailty) were excluded.

## Data extraction

The walking status was evaluated at hospital admission by recording how the patient entered the emergency department. The patient's ability to walk was graded into three categories: 1) can walk unassisted, 2) can walk assisted only, 3) cannot walk. This information was extracted from the trial database. In addition, the following variables were extracted: patient demographics, number of chronic diseases, HIV status, pre-existent malnutrition, presumed etiology and type of infection, danger signs (altered mental state, respiratory distress, systemic hypoperfusion), vital parameters (heart rate, respiratory rate, systolic blood pressure, capillary refill time, temperature), laboratory results (blood sugar, white cell count, hemoglobin levels whenever determined) and the quick Sequential Organ Failure Assessment (qSOFA) score count [7] at hospital admission. The length of hospital stay, total costs of care and survival status at hospital discharge were also extracted.

## Risk factors for death

The qSOFA score ($\geq$2 points) was used to identify adult patients with an increased risk of death as suggested by the Sepsis3 study group [7]. In children ($<$15 years), an increased risk of death was assumed if the child had a suspected or proven infection with any one of the following criteria [8]: body temperature $<36\,°C$, body temperature $>38\,°C$ plus altered mental status, body temperature $>38\,°C$ plus respiratory distress, body temperature $>38\,°C$ plus a history of not feeding, or body temperature $>38\,°C$ plus convulsions.

## Study endpoints and statistical analysis

The primary endpoint of this analysis was to determine the value of the inability to walk unassisted at hospital admission as a predictor of in-hospital mortality. Secondary outcomes were the relationship between the walking status at hospital admission and the presence of risk factors for death or danger signs, vital parameters, length of hospital stay, as well as total costs of care.

The statistical analysis was performed using the PASW statistical software package (IBM SPSS Statistics 20; IBM, Vienna, Austria). Receiver operating characteristic (ROC) analyses

and two-by-two tables were used to determine the sensitivity, specificity, as well as the negative and positive predictive values of the inability to walk unassisted at hospital admission to predict in-hospital death. For this analysis, we grouped patients into subjects who could walk unassisted and those who could not (thus combining the categories can walk assisted only and cannot walk). Descriptive methods were used to evaluate in-hospital mortality of patients with a qSOFA score count $\geq 2$ points as categorized by their walking status at hospital admission. The Chi squared test and an analysis of variance were used to compare binary and linear variables between the three categories of walking status at hospital admission, as appropriate. In view of the fact that 16 comparisons were performed between the three categories, we adjusted the $p$-value for the number of comparisons (Bonferroni correction) and considered $p < 0.003$ to indicate statistical significance. All data are given as median values with interquartile ranges, if not otherwise stated.

## Results

One-thousand-sixty-nine patients were included into this *post hoc* analysis (Table 1). Two-hundred-one (18.8%), 315 (29.5%), and 553 (51.7%) study patients could walk unassisted, walk assisted or not walk at hospital presentation, respectively. The inability to walk unassisted at hospital admission had a low specificity but was 100% sensitive to predict in-hospital mortality (p = 0.00007) (Fig 1). Vice versa, the ability to walk unassisted at hospital admission had a 100% specificity but low sensitivity to predict in-hospital survival. The value of the inability to walk unassisted to predict in-hospital death (AUC ROC, 0.636; CI95%, 0.564–0.707) was comparable to that of the qSOFA score at hospital admission (AUC ROC, 0.622; CI95%, 0.524–0.721).

Fifteen (7.5%), thirty-four (10.8%) and one hundred sixty-seven (30.2%) study patients who could walk unassisted, walk assisted or not walk at hospital admission had a qSOFA score count $\geq 2$ points at hospital admission (Fig 2A), respectively. Hospital mortality differed between these patients when they were categorized by their walking status but this did not reach statistical significance (Fig 2B).

The walking status at hospital admission correlated with an altered mental state, respiratory distress, the qSOFA count, presence of risk factors for death in children, heart rate, respiratory rate, body temperature, white cell count and hemoglobin levels at hospital admission, as well as length of hospital stay, costs of care and survival status at hospital discharge (Table 2).

## Discussion

Rapid identification of patients with serious infection at hospital admission who have the highest risk of death is vitally important if we are to improve their outcome. In a resource limited setting, existing methodologies are not feasible. We believe that there is a need for a simple, binary (Y/N) assessment technique that informs bedside care providers that a patient with serious infection has an increased risk of death.

The results of this *post hoc* analysis suggest that the inability to walk unassisted at hospital was a highly sensitive predictor of in-hospital death in Rwandese patients with a proven or suspected infection. Its value to predict in-hospital death was even comparable to the qSOFA score in our study population. When interpreting this finding, it is important to note that over seventy-five percent [4] of the study population received evidence-based interventions for the management of their underlying infectious disease. These evidence-based interventions included: initiation of antimicrobial therapy; surgical source control; blood glucose measurement and intravenous glucose administration (in cases of altered mental state and hypoglycemia, respectively); placement of patient in recovery position (in cases of unresponsiveness to

**Table 1. Clinical and demographic features of 1,069 patients admitted to the study hospital with suspected infection.**

| Variable | All patients |
|---|---|
| *N* | 1,069 |
| **Age** *(years)* | 20 (7–40) |
| **Age <15 years** *[n (%)]* | 466 (43.6) |
| **Male gender** *[n (%)]* | 486 (45.5) |
| **Chronic diseases** *(n)* | 0 (0–0) |
| **HIV positive** *[n (%)]* | 33 (3.1) |
| **Malnutrition** *[n (%)]* | 7 (0.7) |
| **Duration of symptoms before hospital presentation** *(days)* | 2 (1–5) |
| **Etiology of infection** *[n (%)]* | |
| Bacterial | 205 (28.5) |
| Viral | 86 (8) |
| Fungal | 15 (1.4) |
| Malaria* | 610 (57.1) |
| Parasitic other than malaria | 61 (5.7) |
| **Infectious focus** *[n (%)]* | |
| Meninges | 18 (1.7) |
| Respiratory | 131 (12.3) |
| Abdominal | 13 (1.2) |
| Puerperal | 11 (1) |
| Gastrointestinal | 81 (7.6) |
| Urinary tract | 49 (4.6) |
| Skin/soft tissue | 21 (2) |
| Other | 806 (75.4) |
| **Length of hospital stay** *(days)* | 3 (2–5) |
| **Hospital Mortality** *[n (%)]* | 47 (4.4) |
| **Hospital costs** *(1,000 RWF)* | 29 (18–45) |

*, 61 (5.7%) patients had a co-infection of malaria and another infectious disease.

HIV, human immunodeficiency virus; RWF, Rwandan Francs.

Data are given as median values with interquartile range, if not otherwise indicated.

touch and/or pain); oxygen administration (in cases of respiratory distress); fluid administration (in cases of systemic hypoperfusion) [4]. In our cohort, any further intervention to reduce mortality would have only been meaningful in patients who could not walk unassisted at hospital admission. Therefore, future research efforts to test advanced interventions to reduce mortality in patients with an acute infection in a resource-limited setting should focus on subjects who cannot walk unassisted at hospital admission.

It is highly likely that certain confounding factors other than the infectious disease process (e.g. age, comorbidities) influenced the walking status at hospital admission in our patient population. Indeed, both age and comorbidities have been shown to be significantly associated with mortality in patients with severe infection and sepsis in resource-limited settings [9,10]. However, the aim of our study was not to evaluate the independent association between the (in)ability to walk and hospital survival status, but to identify an easy and reliable triage tool with a high sensitivity to predict in-hospital mortality. Therefore, we chose an unadjusted model to determine the crude value of the walking status to predict in-hospital death.

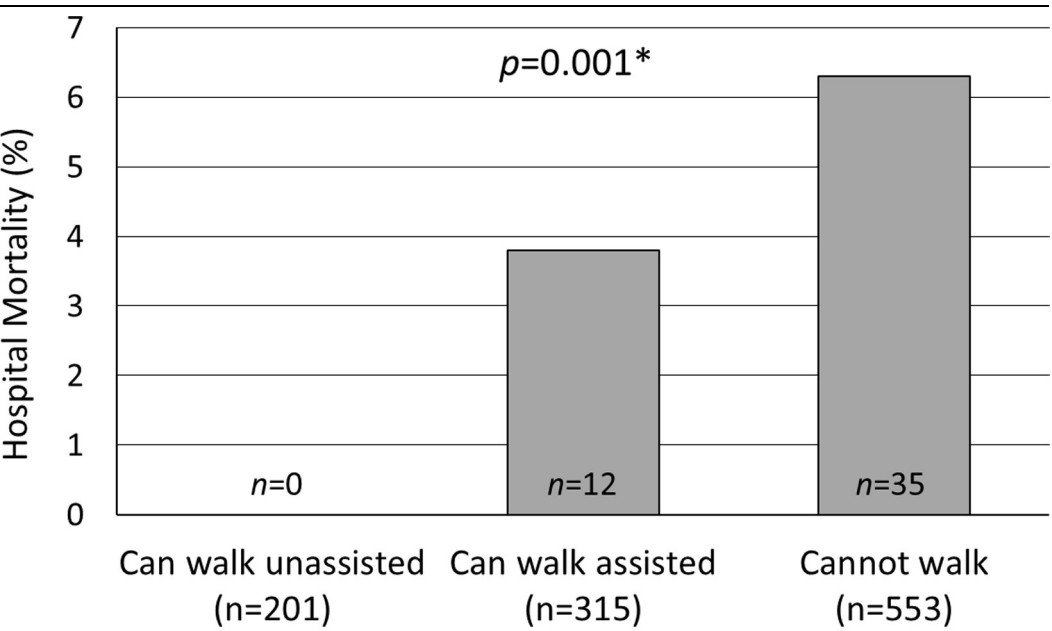

**Fig 1. Hospital survival in patients with proven or suspected infection categorized by their ability to walk.** AUC ROC, area under the receiver operating characteristic curve; PPV, positive predictive value; NPV, negative predictive value; *, Chi$^2$ test.

An interesting finding of our analysis was that the walking status at hospital admission was correlated with the qSOFA score count. The qSOFA score has been shown to be a significant risk factor for sepsis and death both in high- [7] as well as low- and middle-income countries [11]. Notably, the fifteen study patients of our cohort who could walk unassisted to hospital admission and had a qSOFA score ≥2 points all survived. On the other hand, the hospital mortality of study patients with a qSOFA score ≥2 points who could only walk assisted or who could not walk at all was 2.9% and 8.4%, respectively. Although this difference did not reach statistical significance, it is conceivable and biologically plausible that the ability to walk unassisted may be an indicator of the (otherwise difficult to measure) physiologic reserve of a patient suffering from an acute infection.The walking status at hospital admission was also related to several other indicators of disease severity including danger signs, heart rate, respiratory rate and body temperature, white cell count and hemoglobin levels at hospital admission.

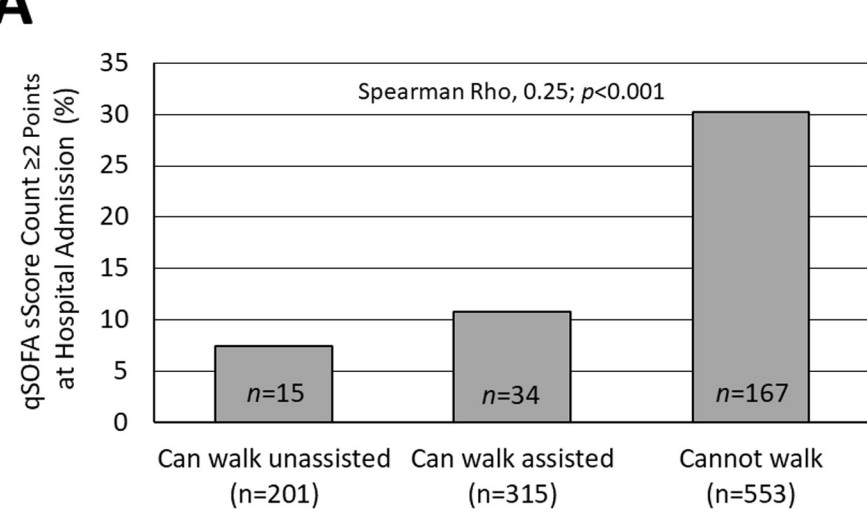

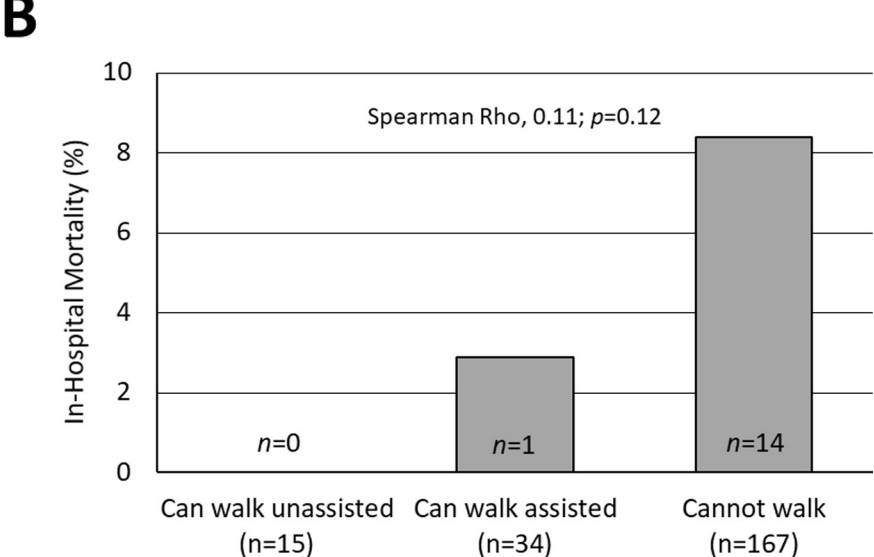

**Fig 2.** Incidence of a qSOFA score count ≥2 points at hospital admission (A) and in-hospital mortality (B) in patients categorized by their ability to walk at hospital admission. *, Chi$^2$ test.

Except for the relationship between the ability to walk and altered mental status, and respiratory rate, most of the relationships observed were vague with correlation coefficients <0.2. This suggests that the walking status at hospital admission is a non-specific indicator of the patient's overall functional condition.

In concordance with our findings, a prospective cohort study from Tanzania reported a clear association between walking status and mortality among medical patients presenting to two hospitals. The ability to walk unaided into the consultation room had a negative predictive value of 97% for hospital mortality [12]. Similarly, the World Health Organization included inability to walk unaided as one of four danger signs to facilitate rapid initiation of empiric anti-tuberculosis treatment among HIV-positive adults with cough [13].

Certain limitations need to be considered when interpreting the results of our analysis. First, we performed multiple comparisons by testing the correlation between 16 parameters

**Table 2. Relationships between the ability to walk at hospital admission and infection source, danger signs, vital parameters, laboratory values at hospital admission, outcome parameters, and costs of care.**

| Variable | Can walk unassisted | Can walk assisted | Cannot walk | Spear-man Rho | *p*-value |
|---|---|---|---|---|---|
| N | 201 | 315 | 553 | | |
| **Danger signs at hospital admission** | | | | | |
| Altered mental state *[n (%)]* | 3 (1.5) | 14 (4.4) | 171 (30.9) | 0.35 | <0.001* |
| Respiratory distress *[n (%)]* | 8 (4) | 20 (6.3) | 77 (13.9) | 0.14 | <0.001* |
| Systemic hypoperfusion *[n (%)]* | 37 (18.4) | 69 (21.9) | 84 (15.2) | -0.06 | 0.06 |
| qSOFA score *(points)* | 0 (0–1) | 0 (0–1) | 1 (0–1.3) | 0.29 | <0.001* |
| Pediatric risk factors for death *[n (%)]* | 7 (3.5) | 19 (6) | 112 (20.3) | 0.15 | 0.001* |
| **Vital parameters at hospital admission** | | | | | |
| Heart rate *(bpm)* | 95 (80–112) | 100 (84–112) | 104 (88–117) | 0.13 | <0.001* |
| Respiratory rate *(bpm)* | 20 (19–22) | 20 (20–22) | 22 (20–26) | 0.31 | <0.001* |
| SBP *(mmHg)* | 112 (99–122) | 108 (98–121) | 107 (94–121) | -0.08 | 0.05 |
| Capillary refill time *(sec)* | 2 (2–2) | 2 (2–3) | 2 (2–3) | 0.05 | 0.11 |
| Temperature *(°C)* | 37 (36.5–37.9) | 37.1 (36.6–38.2) | 37.3 (36.7–38.5) | 0.12 | <0.001* |
| **Laboratory values at hospital admission** | | | | | |
| Blood sugar *(mg/dL)* | 123 (93–161) | 123 (90–156) | 130 (101–166) | 0.11 | 0.12 |
| WCC *(G/L)* | 6.1 (4.6–8.7) | 6.1 (4.1–8.6) | 7.8 (5.3–10.8) | 0.17 | <0.001* |
| Hemoglobin *(g/dL)* | 12.3 (10.6–13.8) | 12.2 (10.1–14) | 11.6 (9.8–13) | -0.13 | <0.001* |
| **Hospital outcome** | | | | | |
| Length of stay *(days)* | 3 (2–5) | 3 (2–6) | 4 (2–6) | 0.11 | 0.001* |
| Costs of care *(1,000 RWF)* | 23.7 (16.2–33.1) | 30.2 (16.9–49.9) | 30.8 (18.6–48.3) | 0.11 | <0.001* |
| Survival *[n (%)]* | 201 (100) | 303 (96.2) | 518 (93.7) | 0.11 | <0.001* |

qSOFA, quick Sequential Organ Failure Assessment; RWF, Rwandan Francs; SBP, systolic blood pressure; WCC, white cell count.

and the ability to walk at hospital admission. Although we adjusted the level of significance for this number of comparisons to $p<0.003$, we cannot exclude the possibility that some relationships were chance findings or reflect collinearity. Second, extrapolation of our results to children who cannot walk unassisted or those with premorbid conditions is not possible. Furthermore, extrapolation to other healthcare facilities, other low-/middle-income countries or even high-income countries may neither be possible as the epidemiology of infection (e.g. malaria), modes of transport and standards of care may differ.

## Conclusions

Our results suggest that the inability to walk unassisted at hospital admission is a highly sensitive predictor of in-hospital mortality in Rwandese patients with a suspected acute infection. The walking status at hospital admission appears to be a crude indicator of disease severity.

## Acknowledgments

Members of the "Sepsis in Resource-Limited Nations" Task Force of the Surviving Sepsis Campaign (in alphabetical order): John I. Baelani, PhD; Danstan Bagenda, PhD; Martin W. Dünser, MD; Joseph C. Farmer, MD, FCCM; Lori A. Harmon, RRT, MBA; Julia T. Hoffman, RN, BSN; Niranjan Kissoon, MD, FCCM; Arthur Kwizera, MD; Mervyn Mer, MD,PhD; Ashok Mudgapalli, PhD; Pierre Mujyarugamba, MSc; Ndidiamaka Musa, MD; Polyphile Ntihinyurwa, MD; Vincent Nyiringabo, MD; Andrew J. Patterson, MD, PhD; Austin M. Porter; Zacharie Rukemba, MD; Hanno Ulmer, PhD; Olivier Urayeneza, MD, FACS.

## Author Contributions

**Conceptualization:** Arthur Kwizera, Olivier Urayeneza, Andrew J. Patterson, Lori Harmon, Joseph C. Farmer, Martin W. Dünser.

**Formal analysis:** Arthur Kwizera, Jens Meier, Martin W. Dünser.

**Funding acquisition:** Olivier Urayeneza, Andrew J. Patterson, Lori Harmon, Joseph C. Farmer, Martin W. Dünser.

**Investigation:** Arthur Kwizera, Olivier Urayeneza, Pierre Mujyarugamba, Andrew J. Patterson, Joseph C. Farmer, Martin W. Dünser.

**Methodology:** Arthur Kwizera, Pierre Mujyarugamba, Jens Meier, Andrew J. Patterson, Joseph C. Farmer, Martin W. Dünser.

**Project administration:** Arthur Kwizera, Pierre Mujyarugamba, Andrew J. Patterson, Lori Harmon, Joseph C. Farmer, Martin W. Dünser.

**Visualization:** Lori Harmon.

**Writing – original draft:** Arthur Kwizera, Jens Meier, Martin W. Dünser.

**Writing – review & editing:** Arthur Kwizera, Olivier Urayeneza, Pierre Mujyarugamba, Jens Meier, Andrew J. Patterson, Lori Harmon, Joseph C. Farmer, Martin W. Dünser.

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
