## [Decision Letter · Decision Letter 0]

23 Oct 2019

PONE-D-19-20519

The ability to walk at hospital admission as a valuable triage tool to predict disease severity and outcome in Rwandese patients with suspected infection

PLOS ONE

Dear Dr. Dünser,

Thank you for submitting your manuscript to PLOS ONE. Your submission has now been peer reviewed by three experts in the field. I agree that the manuscript would benefit from being revised according to the suggestions following and encourage you to do so.

ACADEMIC EDITOR:

My main point is concerning the methodology. I would suggest that you explicit describe how did you define infection (proven or confirmed). Throughout the manuscript you interchangeably use sepsis and infection accompanied by qSOFA with 2 or more points. That is not true as qSOFA did not define sepsis rather than readily identifying those with infection at higher risk of bad outcomes.

Minor issues

Please provide more details about the underlying etiology of infection seen at your study setting. Proportion of bacterial? Viral? Parasitic? Proportion of antibiotic prescription? How many were microbiological confirmed? How many had accompanying organ dysfunction? That information will help the readers understand the context where your investigation was conducted.Page 7. The 95% CI for the PPV/NPV are missing. The prognostic accuracy of the inability to walk at admission should be assessed using the area under the receiver operating characteristic curve (AUROC) and, if possible, compared with qSOFA.Table 1 -  Malaria is not a infection focus but a disease itself (it should move to etiology of infections).Pag 9. I would remove the association between specific disease states and walking status at admission as it did not make much sense.Discussion. I am not sure if measurement of procalcitonin is useful to identify patients and stratify their risk of bad outcomes even in resource-rich settings. 

We would appreciate receiving your revised manuscript by Dec 07 2019 11:59PM. To enhance the reproducibility of your results, we recommend that if applicable you deposit your laboratory protocols in protocols.io, where a protocol can be assigned its own identifier (DOI) such that it can be cited independently in the future. For instructions see: http://journals.plos.org/plosone/s/submission-guidelines#loc-laboratory-protocols

We look forward to receiving your revised manuscript.

Kind regards,

José Moreira, MD, MSc

Academic Editor

PLOS ONE

**Journal Requirements:**

2. We noticed you have some minor occurrence(s) of overlapping text with the following previous publication(s), which needs to be addressed:

doi.org/10.1097/CCM.0000000000003227

In your revision ensure you cite all your sources (including your own works), and quote or rephrase any duplicated text outside the Methods section. Further consideration is dependent on these concerns being addressed.

3. Thank you for stating that “The funders had no role in study design, data collection and analysis, decision to publish, or preparation of the manuscript” in your financial disclosure.

Please also provide the name of the funders of this study (as well as grant numbers if available) in your financial disclosure statement.

**Comments to the Author**

1. Is the manuscript technically sound, and do the data support the conclusions?

Reviewer #1: Yes

Reviewer #2: Partly

Reviewer #3: Yes

2. Has the statistical analysis been performed appropriately and rigorously? 

Reviewer #1: I Don't Know

Reviewer #2: Yes

Reviewer #3: Yes

3. Have the authors made all data underlying the findings in their manuscript fully available?

Reviewer #1: Yes

Reviewer #2: No

Reviewer #3: No

4. Is the manuscript presented in an intelligible fashion and written in standard English?

Reviewer #1: Yes

Reviewer #2: Yes

Reviewer #3: Yes

5. Review Comments to the Author

Reviewer #1: Reviewer comments

1. This is a post hoc analysis of a bigger study. In the methods the authors should clearly state this is a nested study of a bigger study and clarify their patient population and refrain from reporting the patient population which was not part of this analysis.

2. There are some results of the bigger study in the methods. This is confusing I would strongly recommend the sentence be removed.

3. The authors state they included patients who could walk before they sought care for this specific disease. Nothing has been reported of the time frame between the last time the patient walked and this disease.

4. The criteria for suspected infection for the adults which was used in patients with a qSOFA score should be clearly defined the way they have defined the pediatric group.

5. Complete reporting of the sensitivity, specificity and all c-statistics would be helpful.

6. Table 1 can be rearranged and remove the unit column and add those in the first column after the variable name.

7. The title and the results do not connect completely.

8. The figures 1 and 2 can easily be reported in text.

Reviewer #2: The idea behind the study is relevant and novel. I understand why they decided to use the crude value instead of correcting for probable confouders.

The relevance of some of the associations described puzzles me. Association between menigitis or UTI and walking status for example. Walking status is a crude indicator of being sick and I would not try to couple any diagnosis with the walking status. I would ommit these reduntant analysis and keep the message simple. "If a patient with suspected/poven infection cannot walk beware".

In the page 12 they imply that measurement of procalcitonine status is of help in the ED setting. To do what? What is the evidence for this statement?

The authors say that qSOFA > 2 indicate sepsis? (page 13) In my opinion this is not correct. qSOFA.2 indicate bad out comes in sepsis or may be even other diseas states.

In spite of a few methdological shortcomings I think the message is relevant in a resourse limited setting. Therefore I would publish this paper.

Reviewer #3: I read with interest the paper "The ability to walk at hospital admission as a valuable triage tool to predict disease severity and outcome in Rwandese patients with suspected infection". I think it is clearly written and conveys an important message. I have minor comments only, which I have addressed in the attached file. I found the tables and figures unattractive, perhaps the authors could improve this.

6. PLOS authors have the option to publish the peer review history of their article (what does this mean?). If published, this will include your full peer review and any attached files.

Reviewer #1: No

Reviewer #2: No

Reviewer #3: No

---

## [Author Response · Author response to Decision Letter 0]

11 Dec 2019

We thank the editor and reviewers for their critical and thoughtful comments which helped to improve the quality of our manuscript.

Editor:

Specific thanks for the editorial comments to the manuscript itself. All of these have been accepted and included in the revised manuscript.

1. My main point is concerning the methodology. I would suggest that you explicit describe how did you define infection (proven or confirmed). Throughout the manuscript you interchangeably use sepsis and infection accompanied by qSOFA with 2 or more points. That is not true as qSOFA did not define sepsis rather than readily identifying those with infection at higher risk of bad outcomes.

Authors’ response: The author raises an important point. Due to the lack of advanced imaging technology and unavailability of microbiological tests in the hospital laboratory, infection could not be confirmed in the majority of patients (as performed under trial conditions). Pragmatically, the initial suspicion of infection at hospital admission was finally confirmed at hospital discharge based on the clinical response to treatment and exclusion of alternative diagnosis. In the revised manuscript, we clarified this and consistently use the term “suspected acute infection”.

Regarding the diagnosis of sepsis, we agree with the editor that we could not diagnose sepsis in our study population (as suggested by the Sepsis3 definitions), since we did not systematically evaluate organ dysfunction. Therefore, in the revised manuscript, we omitted the term sepsis and only used the term infection. In order to describe patients with an increased risk of death we were more specific and indicated this with a qSOFA Score of 2 points or higher and with the presence of risk factors for death in children. A new paragraph was included into the Materials and Methods section defining these risk factors. The respective paragraph is sub-headed as “Risk Factors for Death” and reads as follows: “The qSOFA score (≥2 points) was used to identify adult patients with an increased risk of death as suggested by the Sepsis3 study group [7]. In children (<15 years), an increased risk of death was assumed if the child had a suspected or proven infection with any one of the following criteria [8]: body temperature <36°C, body temperature >38°C plus altered mental status, body temperature >38°C plus respiratory distress, body temperature >38°C plus a history of not feeding, or body temperature >38°C plus convulsions.”

2. Please provide more details about the underlying etiology of infection seen at your study setting. Proportion of bacterial? Viral? Parasitic? Proportion of antibiotic prescription? How many were microbiological confirmed? How many had accompanying organ dysfunction? That information will help the readers to understand the context where your investigation was conducted.

Authors’ response: In the revised manuscript, Table 1 was relevantly expanded and now includes most of the information suggested by the editor. Some aspects could not be provided as, for example, organ dysfunction could not be systematically evaluated as laboratory tests to do so were not consistently available. This information has also been introduced to the Materials and Methods section of the revised manuscript.

3. Page 7. The 95% CI for the PPV/NPV are missing. The prognostic accuracy of the inability to walk at admission should be assessed using the area under the receiver operating characteristic curve (AUROC) and, if possible, compared with qSOFA.

Authors’ response: All the requested information was added to Figure 1 of the revised manuscript.

Thank you very much for the suggestion to compare the predictive value of the inability to walk unassisted at hospital presentation with that of the qSOFA score. We found that, in our population, the AUC ROC of the inability to walk unassisted was comparable to that of the qSOFA score (0.636 vs. 0.622). This new finding was added to the Results section and addressed in the Discussion section of the revised manuscript.

4. Table 1 – Malaria is not a infection focus but a disease itself (it should move to etiology of infections).

Authors’ response: This was re-arranged. For further details, please refer to our response to the editor’s second comment.

5. Pag 9. I would remove the association between specific disease states and walking status at admission as it did not make much sense.

Authors’ response: In the revised manuscript, the association between specific disease states and walking status at admission were omitted. According changes were made to the Abstract, Materials and Methods, Results, Table 2 and Discussion sections.

6. Discussion. I am not sure if measurement of procalcitonin is useful to identify patients and stratify their risk of bad outcomes even in resource-rich settings.

Authors’ response: The statement referring to procalcitonin was excluded from the Discussion section of the revised manuscript.

Reviewer 1

1. This is a post hoc analysis of a bigger study. In the methods the authors should clearly state this is a nested study of a bigger study and clarify the patient population and refrain from reporting the patient population which was not part of this analysis.

Authors’ response: We agree with this reviewer. Accordingly, the Materials and Methods section of the revised manuscript was largely re-written in order to comply with these points.

2. There are some results of the bigger study in the methods. This is confusing. I strongly recommend the sentence be removed.

Authors’ response: The respective sentences were removed from the revised manuscript.

3. The authors state they included only patients who could walk before they sought care for this specific disease. Nothing has been reported of the time frame between the last time the patient walked and this disease.

Authors’ response: In the Materials and Methods section of the revised manuscript, we clarified that patients who were not able to walk unassisted before this disease episode (e.g. because of young age, neurological conditions, severe chronic disease or frailty) were excluded from the analysis. 

Unfortunately, we did not record the time between the last time the patient walked unassisted and this disease. Therefore, we are sorry that we cannot provide these data. On the other hand, we collected the duration of symptoms before hospital presentation. This new information has been added to Table 1 of the revised manuscript.

4. The criteria for suspected infection for the adults which was used in patients with a qSOFA score should be clearly defined they way they have defined the pediatric group.

Authors’ response: In response to the editor’s comments, we omitted the term sepsis in our revised manuscript. The reason for this was that we did not systematically evaluate organ dysfunction in our study population due to the inconsistent availability of laboratory resources to do so. This information has been included into the Materials and Methods section of the revised manuscript. However, in order to describe patients with an increased risk of death we were more specific and indicated this with a qSOFA Score of 2 points or higher and with the presence of risk factors for death in children. A new paragraph was included into the Materials and Methods section defining these risk factors. The respective paragraph is sub-headed as “Risk Factors for Death” and reads as follows: “The qSOFA score (≥2 points) was used to identify adult patients with an increased risk of death as suggested by the Sepsis3 study group [7]. In children (<15 years), an increased risk of death was assumed if the child had a suspected or proven infection with any one of the following criteria [8]: body temperature <36°C, body temperature >38°C plus altered mental status, body temperature >38°C plus respiratory distress, body temperature >38°C plus a history of not feeding, or body temperature >38°C plus convulsions.”

5. Complete reporting of the sensitivity and specificity and all c-statistics would be helpful.

Authors’ response: This information has been added to Figure 1 of the revised manuscript. 

6. Table 1 can be re-arranged and remove the unit column and add those in the first column after the variable name.

Authors’ response: Changes to Table 1 of the revised manuscript have been made as suggested by this reviewer. Similar changes were made to Table 2 of the revised manuscript.

7. The title and results do not connect completely.

Authors’ response: Thanks for rising this important point. The title was changed and now reads as follows: “The inability to walk unassisted at hospital admission as a valuable triage tool to predict hospital mortality in Rwandese patients with suspected infection”. The short title was accordingly changed to: “Inability to walk as a predictor of mortality in infection”.

8. The figures 1 and 2 can easily be reported in the text.

Authors’ response: We agree with the reviewer that the information presented in Figure 1 and 2 could be included into the text. However, we feel that figures are an important and effective way to present key study results. Therefore, we decided to retain the two figures in the revised manuscript. If the reviewer and/or editor feel strong about omitting the figures and transforming their information into written text, we will revise our decision.

Reviewer 2:

1. The idea behind the study is relevant and novel. I understand why they decided to use the crude value instead of correcting for probable confouders.

The relevance of some of the associations described puzzles me. Association between menigitis or UTI and walking status for example. Walking status is a crude indicator of being sick and I would not try to couple any diagnosis with the walking status. I would ommit these reduntant analysis and keep the message simple. "If a patient with suspected/poven infection cannot walk beware".

Authors’ response: In the revised manuscript, the association between specific disease states and walking status at admission were omitted. According changes were made to the Abstract, Materials and Methods, Results, Table 2 and Discussion sections.

2. In the page 12 they imply that measurement of procalcitonine status is of help in the ED setting. To do what? What is the evidence for this statement?

Authors’ response: The statement referring to procalcitonin was excluded from the Discussion section of the revised manuscript.

3. The authors say that qSOFA > 2 indicate sepsis? (page 13) In my opinion this is not correct. qSOFA.2 indicate bad out comes in sepsis or may be even other diseas states.

Authors’ response: Regarding the diagnosis of sepsis, we agree with the reviewer that we could not diagnose sepsis in our study population (as suggested by the Sepsis3 definitions), since we did not systematically evaluate organ dysfunction. Therefore, in the revised manuscript, we omitted the term sepsis and only used the term infection. In order to describe patients with an increased risk of death we were more specific and indicated this with a qSOFA Score of 2 points or higher and with the presence of risk factors for death in children. A new paragraph was included into the Materials and Methods section defining these risk factors. The respective paragraph is sub-headed as “Risk Factors for Death” and reads as follows: “The qSOFA score (≥2 points) was used to identify adult patients with an increased risk of death as suggested by the Sepsis3 study group [7]. In children (<15 years), an increased risk of death was assumed if the child had a suspected or proven infection with any one of the following criteria [8]: body temperature <36°C, body temperature >38°C plus altered mental status, body temperature >38°C plus respiratory distress, body temperature >38°C plus a history of not feeding, or body temperature >38°C plus convulsions.”

Reviewer 3:

1. I read with interest the paper "The ability to walk at hospital admission as a valuable triage tool to predict disease severity and outcome in Rwandese patients with suspected infection". I think it is clearly written and conveys an important message. I have minor comments only, which I have addressed in the attached file. I found the tables and figures unattractive, perhaps the authors could improve this.

Authors’ response: We thank the reviewer for the kind words and the corrections made to the attached file. We have included all suggestions into the revised version of the manuscript as indicated.

---

## [Editor Report · Decision Letter 1]

28 Jan 2020

The inability to walk unassisted at hospital admission as a valuable triage tool to predict hospital mortality in Rwandese patients with suspected infection

PONE-D-19-20519R1

Dear Dr. Dünser,

Your manuscript has now been formally accepted for publication in Plos One. Please see the essential details concerning the publication process below. Your efforts during the process of revision are acknowledged, and I hope you are also pleased with the final results. 

We appreciate being able to publish your work and look forward to seeing your paper online as soon as possible.

With kind regards,

José Moreira, MD, MSc

Academic Editor

PLOS ONE
---

## [Editor Report · Acceptance letter]

6 Feb 2020

PONE-D-19-20519R1 

The inability to walk unassisted at hospital admission as a valuable triage tool to predict hospital mortality in Rwandese patients with suspected infection 

Dear Dr. Dünser:

I am pleased to inform you that your manuscript has been deemed suitable for publication in PLOS ONE. Congratulations! Your manuscript is now with our production department. 

With kind regards,

on behalf of

Dr. José Moreira 

Academic Editor

PLOS ONE